# Moral Distress in Healthcare Providers Who Take Care of Critical Pediatric Patients throughout Italy—Cultural Adaptation and Validation of the Italian Pediatric Instrument

**DOI:** 10.3390/ijerph19073880

**Published:** 2022-03-24

**Authors:** Chiara Grasso, Davide Massidda, Karolina Zaneta Maslak, Cinzia Favara-Scacco, Francesco Antonio Grasso, Carmela Bencivenga, Valerio Confalone, Elisabetta Lampugnani, Andrea Moscatelli, Marta Somaini, Simonetta Tesoro, Giulia Lamiani, Marinella Astuto

**Affiliations:** 1Department of Anesthesia and Intensive Care, A.O.U. “Policlinico-San Marco”, University of Catania, 95123 Catania, Italy; astmar@tiscali.it; 2Kode, 56122 Pisa, Italy; d.massidda@kode-solutions.net; 3A.O.U. “Policlinico-San Marco” Psychotherapy, 95123 Catania, Italy; maslak.karolina@gmail.com; 4A.O.U. “Policlinico-San Marco” Psychotherapy & LAD ONLUS, 95123 Catania, Italy; cinzia.favara@ladonlus.org; 5Department of Pediatric Anesthesia, Sidra Medicine, Doha P.O. Box 26999, Qatar; fgrasso@sidra.org; 6Pediatric Intensive Care Unit—A.O.R.N. Santobono-Pausilipon, 80129 Napoli, Italy; milena.bencivenga@libero.it; 7Pediatric Intensive Care Unit—Pediatric Emergency Department, Bambino Gesù Children’s Hospital, Institute for Research and Health Care (IRCCS), 00165 Rome, Italy; valerio.confalone@opbg.net; 8Neonatal and Pediatric Intensive Care Unit, Department of Critical Care and Perinatal Medicine, IRCCS Istituto Giannina Gaslini, 16147 Genova, Italy; elisabettalampugnani@gaslini.org (E.L.); andreamoscatelli@gaslini.org (A.M.); 9Department of Anesthesia and Intensive Care Medicine, ASST GOM Niguarda Ca’ Granda, University of Milano Bicocca, 20162 Milano, Italy; marta.somaini@ospedaleniguarda.it; 10Section of Anesthesia, Analgesia, and Intensive Care, Department of Surgical and Biomedical Sciences, University of Perugia, 06129 Perugia, Italy; simonettatesoro@gmail.com; 11Department of Health Sciences, University of Milan, 20133 Milano, Italy; giulia.lamiani@unimi.it

**Keywords:** moral distress, Italian Pediatric Intensive Care, cultural adaptation, Scale Validation, occupational well-being

## Abstract

Background: Although Moral Distress (MD) is a matter of concern within the Pediatric Intensive Care Unit (PICU), there is no validated Italian instrument for measuring the phenomenon in nurses and physicians who care for pediatric patients in Intensive Care. The authors of the Italian Moral Distress Scale-Revised (Italian MDS-R), validated for the adult setting, in 2017, invited further research to evaluate the generalizability of the scale to clinicians working in other fields. Our study aims to reduce this knowledge gap by developing and validating the pediatric version of the Italian MDS-R. Methods: We evaluated the new instrument for construct validity, then we administered it in a multicenter, web-based survey that involved healthcare providers of three PICUs and three adult ICUs admitting children in northern, central, and southern Italy. Finally, we tested it for internal consistency, confirmatory factorial validity, convergent validity, and differences between groups analysis. Results: The 14-item, three-factor model best fit the data. The scale showed good reliability (a = 0.87). Still, it did not correlate with the Emotional Exhaustion and Depersonalization sub-scales of the Maslach Burnout Inventory (MBI) or with the 2-item Connor-Davidson Resilience Scale (CD-RISC 2) or the Satisfaction with Life Scale (SWLS). A mild correlation was found between the *Italian Pediatric MDS-R* score and intention to resign from the job. No correlation was found between MD and years of experience. Females, nurses, and clinicians who cared for COVID-19 patients had a higher MD score. Conclusions: The *Italian Pediatric MDS-R* is a valid and reliable instrument for measuring MD among Italian health workers who care for critically ill children. Further research would be helpful in better investigating its applicability to the heterogeneous scenario of Italian Pediatric Critical Care Medicine.

## 1. Introduction

Working in a Pediatric Intensive Care Unit (PICU) is associated with a higher risk of Moral Distress (MD) when compared with working in Neonatal and Adult Intensive Care Units (ICUs), or other Pediatric Wards [1,2]. Moral Distress was identified for the first time in 1984 as the painful psychological disequilibrium resulting from making a clear moral judgment about what action one should take, but being unable to act accordingly, due to social, institutional, or environmental constraints [3]. On the one hand, in some cases, it could, to a certain extent, drive corrections and progress in the workplace. On the other hand, it can lead to compromised personal integrity, burnout, decisions to resign from the job or from the high-intensity unit. It can also have negative effects on patient care, since it can cause healthcare professionals to restrain their emotional connection with patients and families [4]. Therefore, there is broad agreement that MD has a detrimental effect on both a personal and professional level [5,6], and many studies have focused on it, ranging from descriptive correlational to experimental research. However, most of these articles lack scientific rigor and present significant threats to internal and external validity; thus, measurable and methodologically rigorous studies are currently requested to build evidence-based literature in this field [7].

The most common assessment instrument for MD is the Moral Distress Scale-Revised (MDS-R) [2,8,9,10] which has been specifically adapted to the pediatric setting by Hamric et al. in 2012 [8]. Lamiani and colleagues, in 2017, validated the Italian Moral Distress Scale-Revised (Italian MDS-R) for Italian healthcare providers in adult settings [11]. To our knowledge, there is no validated Italian instrument to measure MD in both nurses and physicians who take care of pediatric patients in Intensive Care Units. While recent studies have analyzed the phenomenon among PICUs worldwide [1,2,12,13,14], only two relate to Europe [15,16]. Of these, the Italian study focuses only on PICUs located in the north of Italy [15]. However, the Italian healthcare system presents differences with an infant mortality rate in the south significantly higher than in the north (North-East 2.24 vs. Islands 3.79—ISTAT data 2017), and an uneven PICU distribution. As a result, approximately 50% of Italian critical pediatric patients are still hospitalized in adult ICUs [17]. Lamiani, in 2017, stated that, since her study concerned a sample of adult ICU professionals in a single region of northern Italy, “the generalizability of the scale to clinicians working in other settings should be confirmed” [11].

Our research aims to reduce this knowledge gap by developing and validating a pediatric version of the Italian MDS-R, the *Italian Pediatric MDS-R*, representing a more reliable and accurate measurement of the phenomenon throughout the country. The new instrument is suitable for measuring the MD in those who care for critically ill pediatric patients, both in Italian PICUs and adult ICUs admitting children, and in northern, central, and southern Italy. We evaluated construct validity by pre-testing and cognitive interviewing. Then, we administered the *Italian Pediatric MDS-R* in a multicenter, cross-cutting, web-based survey. Finally, we assessed the reliability and validity of the new instrument through internal consistency, factorial and convergent validity, and differences between groups analysis.

## 2. Materials and Methods

To adapt the psychometric instrument to a different setting, we followed the three-phase, nine-step approach proposed by Boateng in 2018 [18], as shown in Figure 1.

The MoDiPerSaPerCI (**Mo**ral **Di**stress nel **Per**sonale **Sa**nitario che si occupa dei **P**azienti **Pe**diat**r**ici **C**ritici in **I**talia) research group conducted the study. The research group involves physicians and nurses from three PICUs and three Adult ICUs admitting children in northern, central, and southern Italy and Italian psychotherapists and psychologists. All the steps were traced through appropriate documentation and shared with the MoDiPerSaPerCI group.

### 2.1. Ethics

The Ethics Committee of the Coordinator Center, the Department of Anesthesia and Intensive Care, Presidio “G. Rodolico”, A.O.U. “Policlinico-San Marco”, Catania (prot n. 46755, 05.12.2020–n. 113/2020/PO Reg. Par. CE) and the local Ethics Committees of all the participant centers approved the study.

We obtained, from Epstein and Lamiani, permission to use, respectively, the MDS-R and the Italian MDS-R scales as models for the development of the *Pediatric Italian MDS-R*. We also informed the other authors of the Italian scales [19,20] that we would use their instruments as a comparison during the content validity assessment section. Finally, we obtained permission to use either the 2-item Connor-Davidson Resilience Scale (CD-RISC 2) [21] or the Italian version of the Satisfaction with Life Scale (SWLS) (5 items) [22] as convergent validity instruments.

Consent to participate in the study and to process personal data has been explicitly given, in written form, by each enrolled person. Participation in the survey was anonymous and voluntary. The platform used was compliant with the GDPR (General Data Protection Regulation)–UE 2016/679, and it does not allow tracing of the participants’ identities (https://www.surveymonkey.com/mp/privacy/ accessed on 31 October 2021).

### 2.2. First Expert Panel and Content Validity Evaluation Conducted by Experts

The first expert committee was composed of eight members, including specialists in Pediatric Critical Care Medicine, language experts (Italian and English), professionals with experience in developing and translating psychometric instruments, and the developer of the original Italian MDS-R. The participants received scientific materials in advance that helped them to be consistent with previous translations [11,15,20]. Following the guidelines for cross-cultural adaptation of self-report measures [23,24], the expert panel confirmed the lack of instruments to measure the MD in the Italian Pediatric Critical Care setting. They also reached a consensus on both taking the 14-item Italian MDS-R as the primary reference instrument and assuming the MD of Italian healthcare professionals, who take care of critically ill children, as the domain of the new instrument. They read the questionnaire and compared the Italian MDS-R, the original English version (MDS-R), and any previous Italian-validated MD scales. Finally, they identified keywords and “sensitive” items that deserved to be explored during the pre-test and cognitive interview phases and made “preliminary” changes. They made all their decisions with the intention of obtaining semantic, idiomatic, experiential, and conceptual equivalence between the original Italian MDS-R, previous versions of the MD scale, and the new pediatric one [24].

#### 2.2.1. Choice of the Instrument

The instrument to measure MD is the Moral Distress Scale (MDS), the first version of which was published by Corley in 2001 [9]. In 2010 Epstein and Hamric modified Corley’s scale, giving rise to a shorter 21-item, three-factor instrument, the Moral Distress Scale-Revised (MDS-R). Their scale consisted of six versions to make it usable for all workers in the health sector, both in adult and pediatric settings [5]. The MDS-R is currently the most diffused instrument for the study of MD in Intensive Care workers, having demonstrated good reliability and validity [10]. Since other root causes of MD have been highlighted over the last few years, the same authors, in 2020, produced a new version of the instrument, the Measure of Moral Distress for Healthcare Professionals (MMD-HP). The latter has 27 items and four factors (system level, patient level, team level personal integrity, team level team interaction) and is unique for all healthcare providers. Still, it has never been translated into Italian [25].

Despite many available Italian scales on MD, only a few have been tested for validity. In the adult setting, Badolamenti, in 2017, validated an 11-item, two-factor version for Italian nurses in all specialties [20]. In the same year, Lamiani produced the Italian Moral Distress Scale-Revised (Italian MDS-R) for all healthcare providers who care for adult patients in an ICU setting. This instrument resulted from the Italian translation of the MDS-R and its in-depth cultural adaptation and validation. This scale grouped the three versions for the adult setting of Hamric’s MDS-R in a single 14-item scale, where items 1, 10, 11, 12, 13, 17, and 21 have been removed, and where four sub-scales can be recognized (Futile Care, Ethical Misconduct, Deceptive Communication, Poor Teamwork) [11]. In the Italian pediatric setting, Lazzarin translated Corley’s first version of the MDS, giving rise to the Moral Distress Scale Pediatric Version (MDSPV), which demonstrated high reliability (Cronbach’s alpha of 0.959) when used among nurses on pediatric oncology wards [19].

Although the MMD-HP is the most recent and comprehensive scale [25], we used Lamiani’s 14-item Italian MDS-R as the main reference [11]. The reasons for this decision are that it was already validated for Italian adult intensive care settings and because it derived directly from the MDS-R, which is, currently, the most used instrument in MD research worldwide [2,5,6,7,8,9,10].

We developed the *Italian Pediatric MDS-R* involving the author who validated the adult one [11]. The new scale differs from the Italian MDS-R in the presence of the term “parents” instead of “family members” in items 1, 2, 4, 11; in the presence of the word “child” instead of “patient” in items 3, 6, 9, 11, 12; in the presence of the term “parents” instead of “patients” in item 14.

#### 2.2.2. Scoring System of the Italian Pediatric MDS

The MDS-R requires coding the ratings on the intensity and frequency scales using five integers between 0 and 4. The frequency score ranges from “never” (0 points) to “very often” (4 points). The intensity score varies from “none” (0 points) to “very unpleasant” (4 points), and is assigned to each item, regardless of whether the situation described ever occurred in the interviewee’s experience. If that specific situation has never happened (frequency score 0), the respondent is asked to rate the intensity of the disturbance that would cause that particular situation should it occur. Each item score is the product of the frequency and the intensity ratings and can assume a value between 0 and 16. The total score of the *Italian Pediatric MDS-R* results from the sum of the composite scores obtained on all items (range 0–224) divided by the number of items (resulting in a range 0–16).

Not all the values included in the range 0–16 have the same opportunity to occur since not all can be obtained by combining the by-product of the two original number sequences: the resulting score can assume only ten values (instead of 17). Furthermore, many intensity and frequency products can result in the same value, so there are values with an a priori higher probability of occurring than others. For example, the value 0 results from as many as 9 out of 25 combinations and has a 36% chance of arising. This feature of the scoring system can easily lead to asymmetric distributions.

### 2.3. Pre-Testing and Cognitive Interviewing. Content Validity Evaluation by the Target Population

To evaluate the face and cultural validity of the scale, we performed pre-testing of the instrument followed by cognitive interviewing of seven nurses and seven physicians who work in ICUs throughout Italy. We performed purposive sampling following a parallel, or non-interlocked, method, as recommended by Collins [26]. We set the following as selection criteria (quotas): role (physicians and nurses), age (<50 years old and ≥50 years old), hospital characteristics (PICU and Adult ICU), and location (northern, central, and southern Italy), aiming at a representative sample of the population of interest. After direct recruitment, this study phase took place on a web platform for three weeks and was video recorded. The Cognitive Interview was conducted by two psychotherapists, following the 4-stage model devised by Tourangeau in 1984 [27] and Collins’ method [26]. It followed a semi-structured protocol previously agreed between the interviewers and the researchers who created the questionnaire, one of whom also acted as an observer. Comfortability in answering was scored using an interviewee self-assessment scale from 0: “not at all comfortable” to 4: “completely comfortable”. Comprehension was evaluated using a score from 0: “no understanding” to 1: “total understanding”, rated by the author of the questionnaire and the interviewer. At the end of this section, a descriptive and explanatory analysis report was prepared and then discussed during the second expert panel stage to make further adjustments to the new instrument.

### 2.4. Survey Administration

To further develop the new scale, we proposed a multicenter, cross-cutting, no-profit, web-based survey among the healthcare providers who take care of critically ill pediatric patients in three PICUs (UOSD Terapia Intensiva Neonatale e Pediatrica–IRCCS Istituto Giannina Gaslini, Genova; Area Rossa–Dipartimento di Emergenza e Accettazione–IRCCS Ospedale Pediatrico Bambino Gesù, Roma; Rianimazione Pediatrica A.O.R.N. Santobono-Pausilipon, Napoli) and in three Adult ICUs admitting children (SC Anestesia e Rianimazione I, Azienda Ospedaliera Ca′ Granda Niguarda, Milano; Unità di Terapia Intensiva. Azienda Ospedaliera di Perugia, Perugia; U.O. Terapia Intensiva, Presidio “G. Rodolico”, A.O.U. “Policlinico-San Marco”, Catania) located in northern, central and southern Italy.

Participants were eligible if they were physicians or nurses who have taken care of at least one critically ill pediatric patient (age at the admission > 28 days and ≤16 years old) in the last 12 months and have worked in the enrolled Unit for at least three months. In addition, they had to be willing to answer the online questionnaire voluntarily and anonymously. Providers who were off work for ≥30 days, and those who did not give their consent, were excluded.

The survey provided for anonymous and voluntary participation and was accessible via a specific web link set up as anonymous data collection on the SurveyMonkey online platform. The weblink was sent to the enrolled population by the local referent of each Center through the institutional email address. We asked participants to answer the questionnaire referring exclusively to their work with critically ill pediatric patients. Recruitment in each Center began on the first day following the approval of the local Ethics Committee and lasted six weeks over a total period of 6 months. No time limits were set for the interviewee when completing the questionnaire; however, the estimated average response time was 15 min. The study coordinator provided a weekly reminder during the recruitment period.

For survey purposes, the *Italian Pediatric MDS-R* was incorporated in a 55-item questionnaire which comprised the following sections:Sociodemographic information: age, gender, parenthood.Workplace information: years of experience, role, professional background, hospital characteristics and location, and involvement in treating COVID-19 patients.


*Italian Pediatric MDS-R*
Self-reported perceived psychological status about current stressors in private life, MD level, the effect of working during a pandemic on the worker’s own emotivity, hospital resources and support, and intention to resign from the job.Burnout, according to two subscales of the Maslach Burnout Inventory (MBI)–Italian Validated Version [28], was rated on a Likert 7-point scale (0–6). Specifically, Depersonalization (5 items) with a score of ≤3 was rated low; 4–8 medium and ≥9 as high depersonalization; and Emotional Exhaustion (9 items) with score ≤ 14 was rated low; 15–23 medium and ≥24 as high emotional exhaustion.Resilience was measured using the 2-item Connor-Davidson Resilience Scale (CD-RISC 2): items cd_risc1 and cd_risc2 investigate the ability to adapt to changes and recover from difficulties, respectively. The instrument is rated on a 5-point scale (range 0–4), with higher scores reflecting increased resilience [21].Satisfaction with life according to the Italian version of the Satisfaction with Life Scale (SWLS) (5 items) [22] was rated on a 7-point scale (range 1–7) with higher scores reflecting higher satisfaction with life.Participation in potentially relevant mitigating activities.


### 2.5. Sample Selection

Surveys completed more than once that had two or more missing answers in the *Italian Pediatric MDS-R*, or any sub-scale of the MBI, were excluded. Furthermore, inferential analyses restricted the sample to participants showing variability between items, both in intensity and frequency of scale of the *Italian Pediatric MDS-R*.

### 2.6. Hypotheses for Validity Testing

To analyze the new Italian Pediatric instrument for validity, we reviewed available literature on MD and generated three hypotheses. First, we hypothesized that nurses would have higher MD than physicians. This scenario has been described as a stable trait of the MD scales and is likely related to the less powerful position nurses have in the units [8,14,25,29,30,31]. Second, we would expect healthcare providers considering leaving their position to have higher MD levels than the others, as already demonstrated in many previous studies [8,25,29,31]. Third, we hypothesized that the MD score correlated with burnout measured using the Depersonalization and Emotional Exhaustion subscales of the MBI [14,20,32,33].

### 2.7. Statistical Analysis

We inspected the distribution along with the intensity and frequency scales. Further, the composite score for each item was calculated by multiplying the intensity and the frequency scores, obtaining a single value for each subject by item.

The primary analyses aimed to study the dimensionality of the *Italian Pediatric MDS-R* and its internal consistency. We assumed the factorial structure identified by Lamiani et al. [11] as reference. Lamiani’s instrument consists of the following four dimensions:Futile Care (FC), items: 2, 3, 6;Ethical Misconduct (EM), items: 5, 7, 9, 10, 11;Deceptive Communication (DC), items: 1, 4, 14;Poor Teamwork (PT), items: 8, 12, 13.

We assessed the internal consistency of each scale using Cronbach’s Alpha (target: α ≥ 0.70). Furthermore, we inspected the correlations between, and within, factors.

A Confirmatory Factor Analysis (CFA) was run to validate the latent structure of the instrument. The model was fitted with free covariance parameters between the latent factors, conforming to Lamiani’s findings. We identified the following potential models to study and compare with references from exploratory analyses:Unidimensional: A model considering a single latent factor.Lamiani et al.: Model identified by Lamiani and colleagues, described above.Modified-1 Lamiani et al.: A modified version of the model by Lamiani et al., where item 14 loads on the factor PT instead of the factor DC.Modified-2 Lamiani et al.: Another modified version of the model by Lamiani et al., where item 14 loads both on the PT and the DC factors.Three-factor: A model obtained by dissolving the scale DC and assigning its three items to the other scales (item 1 to FC, item 4 to EM, item 14 to PT). 

The preliminary exploratory study identified the last three models since they accommodate the correlation structure found in our sample.

Model comparison was performed considering various fit indices:Akaike Information Criterion (AIC, threshold: As small as possible);Bayesian Information Criterion (BIC, threshold: As small as possible); Comparative Fit Index (CFI, acceptable for values ≥ 0.95);Tucker-Lewis index (TLI, acceptable for values ≥ 0.95);Root Mean Square Error of Approximation (RMSEA, acceptable for values ≤ 0.06); Weighted Root Mean Square Residual (WRMR, acceptable for values < 1).

After the model fitting, we studied the correlations between factors. Afterwards, we calculated a total score by averaging all the item-level scores. Then we used this resulting score to study the relation of the instrument with external criteria, such as the subscales of the MBI, the CD-RISC 2, the SWLS, and other responses provided by participants to other sections of the survey. We applied Pearson and polyserial correlation indices to compare interval scores and Wilcoxon/Mann-Whitney tests to perform group comparisons.

We performed the statistical analyses using the R software version 4.0.5 with the package *lavaan* 0.6.8 and fitted models utilizing the robust MLMV estimator because of the skewness of the composite scores.

## 3. Results

### 3.1. Cognitive Interviewing

The average duration of the interviews was 1 h and 19 min (min 1, max 3 h). None of the 14 participants experienced difficulty using the Survey Monkey interface on computers, phones, or tablets. Their demographics are shown in Table 1. Respondents were almost always at ease in answering (mean 3.4, median 3.4, SD ± 0.3) except for a few cases in which the recall of the evoked clinical experience, or the need to identify a typical situation, made them uncomfortable. Overall, the answers had normal variability, and no significant differences were identified between physicians and nurses. Comprehension was on average good (mean 0.88, median 0.92, SD ± 0.13). Only the item “Follow the family’s request not to discuss death with a dying child who asks about dying” rated 0.7 and was often misunderstood to mean that the interlocutor was the parent and not the child. Only in a few cases (the introduction section and items 3, 4, 9, and 11) did we estimate that the final questionnaire data’s significance and representativeness could be influenced by ambiguity, lack of clarity, or cultural differences. Therefore, we have made the changes shown in Appendix A.

### 3.2. Survey

The survey was answered 231 times over a 6-month period, between April and October 2021. All respondents gave their informed consent. Thirty compilations were excluded due to insufficient data because they had omitted answers equaling at least half of the questions proposed by the questionnaire (39 out of 78). Therefore, we analyzed 201 responses for a response rate of 82.7%. None of them claimed to have already answered the survey. Then we also excluded the following: three responses because they had one or more missing answers for each MBI sub-scale, 15 which stated they had not treated critically ill pediatric patients in the last 12 months, and one because it met both exclusion criteria. Following the selection operations, 182 participants remained for the data analysis.

Most respondents were nurses (63.7%), female (67.6%), and had between 10 and 19 years of experience (26.9%). However, a large second group (23.1%) had between 1 and 4 years of experience in critical care. The demographic data are shown in Table 1.

#### 3.2.1. Italian Pediatric MDS-R

We did not include five respondents who presented constant ceiling or floor responses on the frequency or intensity scale in the validation study. As a result, we conducted our analyses on 177 participants.

We performed a preliminary analysis to explore the mean location of the sample along with the response criteria (i.e., intensity and frequency) (Table 2). After coding responses using integers between 0 and 4, we calculated the mean by the subject of all item scores separately for the intensity and frequency scores. Then we placed results in a scatterplot (Figure 2) which shows that, generally, items elicited unpleasant sensations (high intensity), but they were rarely experienced (low frequencies).

We analyzed the correlation between intensity and frequency scores using both the Pearson and the polychoric index for each item, and we found weak values (*r* < 0.3) indicating independence between the two response criteria (Appendix A). Furthermore, the observed distributions of the item scores showed heavy asymmetries due to a high frequency of zeros, both for intensity and frequency (Appendix A). 

#### 3.2.2. Reliability

We calculated the items’ composite scores. Figure 3 represents Pearson correlations where the instrument is visualized as a network. The maximum correlation value is *r* = 0.63 (for the full correlation matrix, see Appendix A). Figure 3 shows that the items of the Futile Care (FC), Ethical Misconduct (EM), and Poor Teamwork (PT) scales are close to each other. This finding matches what we expected, assuming the factorial structure of Lamiani et al. Conversely, a relevant separation was found between the items on the Deceptive Communication (DC) scale where item 14 (“Ignore situations in which parents have not been given adequate information to ensure informed consent”) appears better linked to the PT scale, while items 1 (“Witness healthcare providers giving “false hope” to parents”), and 4 (“Follow the family’s request not to discuss death with a dying child who asks about dying”) connected to both FC and EM factors. Based on their semantics, we then relocated items 1 and 4 to FC and EM.

The overall Cronbach alpha was 0.87, indicating excellent internal consistency. However, we need caution to judge this value because of distributional asymmetries. The alpha values of the individual scales were: 0.81 for FC, 0.74 for EM, 0.47 for DC, and 0.72 for PT. Therefore, as already seen from the correlation analyses, the DC scale has low internal consistency, while the other scales showed good indices (>0.7).

#### 3.2.3. Factorial Validity

The CFA shows the models with the best fit are Modified-2 Lamiani et al., followed by the Three-Factor Model. The latter was the best solution because it showed the lowest AIC and BIC, has CFI and TLI above 0.9, RMSEA close to 0.6, SRMR less than 0.8, and WRMR just over 1. Moreover, the three-factor model has the simplest structure. It does not deny the original factorial form, but simplifies it by dissolving a factor of dubious consistency, and does not present problems in the variance-covariance matrix of the latent variables (Appendix A). We adapted a version for each model with complete independence of the latent factors, but this structure worsened the indices. Finally, starting from the three-factor model, we conducted a multi-group analysis considering profession and gender identity as stratification variables, but we did not observe any improvement. The model, therefore, seems to be invariant for what concerns these variables. The model is visualized in Figure 4.

#### 3.2.4. Correlation Analysis and Factor Scores

The CFA led to estimating FC, EM, and PT factor scores. This statistical calculation differs from daily practice, where the factor scores are usually obtained as a simple sum (or average) of the raw scores of the items. For this reason, we verified the correlations between the factor scores estimated by the model and the factor scores calculated by a simple sum of the raw items. The results presented very high correlations, enough to justify the practical choice: *r* = 0.981 for FC, *r* = 0.954 for EM, and *r* = 0.973 for PT.

We calculated a total score as the sum of each item’s intensity and frequency product. Then, we calculated the correlations of this total score with each factor score, resulting in *r* = 0.833 for FC, *r* = 0.903 for EM, and *r* = 0.699 for PT. Raw-scored factors presented modest to moderate correlation:FC and EM: *r* = 0.657;FC and PT: *r* = 0.304;EM and PT: *r* = 0.516.

#### 3.2.5. Convergent Validity Test

The total score of the *Italian Pediatric MDS-R* shows low polyserial correlation with CDRISC 2 scales (cd_risc1: *r* = −0.22; cd_risc2 *r* = −0.01) and SWLS scales (values within *r* = 0.1). Moreover, no relevant correlations were found with the MBI scales Emotional Exhaustion (EE; *r* = 0.1509) and Depersonalization (DP; *r* = 0.119).

However, regarding relevance of the MBI, we deepened the correlation study using a visual inspection and considering the single factors of the *Italian Pediatric MDS-R* (Figure 5). Mainly referring to the relation with the MBI-DP scale (Figure 5b, red), plots show a fan-like structure where it seems possible to identify three groups: subjects with low moral distress and no burnout; subjects with high moral distress and low burnout; subjects with high moral distress and high burnout.

The *Italian Pediatric MDS-R* has a mild correlation with the self-assessment of Moral Distress (*r* = 0.235). It also appears connected to intention to resign due to MD (Wilcoxon’s *W* = 1233.5, *p* = 0.036): median score of subjects intending to resign is 66 (*n* = 22), while the median score of the subjects not intending to resign is 53 (*n* = 155). We did not find any relevant correlation between the MD score and the number of years of experience (*r* = 0.073), overall hospital support, and potentially mitigating activities, except for debriefing (*r* = −0.26) and ethical meetings (*r* = −0.238). Self-assessment of the effect of working during a pandemic on workers’ emotions did not correlate with MD score (*r* = 0.194). Still, healthcare providers who cared for COVID-19 patients had a higher MD score (Wincoxon’s *W* = 1937, *p* = 0.04). Regarding socio-demographic variables (Table 3), we found a higher MD score in females than males (Wilcoxon’s *W* = 4396.5, *p* < 0.01), and in nurses than physicians (Wilcoxon’s *W* = 4381, *p* = 0.019). Healthcare providers in northern Italy (Kruskal-Wallis *chi-squared* = 14.234, *p* < 0.001) and those who care for children in adult ICUs (Wilcoxon’s *W* = 1903, *p* < 0.001) scored higher in the *Italian Pediatric MDS-R*.

## 4. Discussion

Although MD in healthcare providers is a matter of concern in Italy, to our knowledge, there is a lack of existing validated Italian instruments for measuring the phenomenon in nurses and physicians who care for pediatric patients in an Intensive Care setting. Our study adapted the Italian MDS-R and developed a psychometric scale to measure MD among Italian healthcare workers who take care of critically ill pediatric patients, in both PICUs and adult ICUs admitting children.

This research reveals promising evidence of the *Italian Pediatric MDS-R* reliability and validity (Cronbach a 0.87); however, differing from Lamiani’s research, our analysis found the three-factor model is more appropriate than the four-factor one. Moreover, our factorial consistency analysis indicated that the DC scale has poor internal consistency with a much lower Cronbach than the other three (DC 0.47 vs. FC 0.81, EM 0.74, PT 0.72). In addition, item 14 (“Ignore situations in which parents have not been given adequate information to ensure informed consent”) is located far from the other two DC-belonging items, which was shown to be well linked to both FC and EM factors instead. The “destiny” of the DC scale is undoubtedly a point requiring further in-depth analysis. However, since this paper aims to direct future investigations on a comprehensive and more representative sample, we preferred not to exclude the items of the DC scale, awaiting a second stage of the study. Thus, given our CFA results, we dissolved the DC factor by assigning its three items to the other scales (item 1 to FC, item 4 to the EM, item 14 to the PT) regarding their semantics. In the analysis conducted by Lamiani, even though item 14 belonged to the DC scale, it also showed a significant loading on the EM factor [11]. In our analysis, however, this item seems better connected with all three items of the PT factor. Our finding correlates with Epstein’s recent study on the MMD-HP scale, where the corresponding item is part of poor teamwork root causes which specifically pertain to problems in the team’s interaction with families and patients [25]. The connection between informed consent and poor teamwork is probably due to the specificity of the ICU setting, where patients are often in life-threatening conditions and undergo a variety of risky and unfamiliar procedures for days or weeks. This situation can overwhelm stressed families and be perceived as time consuming by ICU doctors [34,35]. Regarding item 1 (“Witness healthcare providers giving “false hope” to parents”), we found it better fits the FC factor. The complex concept of giving “false hope” in healthcare coincides with situations where healthcare workers provide futile, non-beneficial, or even harmful treatments or other actions despite the fact that this “offers no reasonable hope of recovery or improvement” [36,37]. Moreover, it also leads providers to deviate unreasonably from the standard of care [38]. We then assigned item 4 (“Follow the family’s request not to discuss death with a dying child who asks about dying”) to the EM factor because the situation it describes currently constitutes an ethical dilemma [39,40].

We also found that the three-factor model better mirrors the three-level root causes initially identified in MD: patient, system, and the team, as well as the idea that some of these operate at more levels [9,41]. Moreover, even though we also assessed the suitability of the unidimensional model during this study, we found that a multifactorial model better expressed the complexity of the ICU environment. It offers a more detailed view of the causes of MD and allows for more appropriate interventions [11,25,42].

Our study is the first that verified the correlations between the factor scores estimated by the CFA model and the one calculated by summing the items, without reapplying the weighting structure identified by CFA, and it showed a strong correlation between them. This result supports the clinical practice of scoring the instrument by summing item-level values.

Regarding validity testing, two out of three hypotheses strongly supported the validity of the *Italian Pediatric MDS-R*. First, in our sample, in coherence with most MD studies [1,8,10,25,29], nurses have a higher MD score than physicians. Interestingly, this differs from Lamiani’s research which found no differences in age, sex, role, and experience [11]. We believe this might be due to our sample’s social and cultural characteristics and requires further studies on the Italian population to be clarified. Second, clinicians who intend to leave their position due to MD had a higher score at the *Italian Pediatric MDS-R*. This effect of MD is recognized by Lamiani’s study and other previous ones as one of its wasteful and costly effects on the healthcare system [25,29,43,44]. Similarly, we found that 22 out of 182 (12.1%) respondents were considering leaving their current job due to MD, 59 (32.4%) thought about leaving their position at least once in their life for the same reason, and 6 (3.3%) have already changed jobs. In contrast to available data [1,5,20,33,45], our study did not find a significant correlation between MD score and Emotional Exhaustion (EE) and Depersonalization (DP) sub-scales of the MBI. However, other researchers who found a linkage between MD and burnout reported low-intensity correlations, although higher than ours [14,33]. Therefore, as we consider MD a psychological construct subject to high variability, we believe that this aspect deems further research with a larger sample size necessary before drawing any conclusions. 

We also extended our study of MD-burnout relationships with a visual inspection, resulting in a fan-like structure where a correlation, mainly for low scores, can be recognized (Figure 5). Considering limitations linked to the small sample size and resulting high variability, it seems possible to identify the following three groups, mainly regarding the relation with the MBI-DP scale: two in which MD and Burnout correlate for low and high values respectively, and a third group in which high MD values are associated with low burnout (Figure 5b, red). We believe the latter group is worth being further analyzed to better explain this phenomenon and the effect that support, and mitigating activities, can have [4]. 

Finally, our data show that the less resilient providers had a higher MD. However, this did not reach statistical significance on the convergent validity test. Many studies recently analyzed the correlation between resilience measured using the CD-RISC 2 scale and MD, with diverging results [46,47]. In addition, the latest MD mitigation strategies have development of moral resilience as a goal. They aim not to eliminate the MD from the ICU environment but rather to maintain it as a positive state of activation (eustress) and control its progression towards distress, burnout, and other adverse effects, such as intention to leave the job [4,48]. 

This initial testing of the *Italian Pediatric MDS-R* provided evidence of reliability and validity, demonstrating that the new instrument functions correspondingly with the other scales from which it is derived [8,11,25]. However, additional studies are needed to enhance these findings and draw conclusions. Unlike most recent studies on cultural adaptation of MD measurement instruments [49,50,51,52], which only tested reliability, our one includes in-depth validation analysis with dimensionality and validity tests. We deem it proof to consider the new scale an appropriate measure of MD and a valuable base for further research efforts. Furthermore, our data demonstrated that MD is a relevant and costly problem among Italian pediatric critical care settings with differences between PICUs and adult ICUs admitting children and between north, central, and south Italy. We consider this finding to be worth further exploration. We provide a fundamental instrument for further evidence-based research, monitoring, and interventions to guarantee patients’ safety and healthcare professionals’ well-being. 

## 5. Limitations

Our study has some limitations:

Firstly, the sample size is relatively small, resulting in high variability and potentially low accuracy of the results. Therefore, our data should be interpreted with caution, and further studies should investigate this topic.

Secondly, voluntary participation in the survey can cause bias due to the self-selection of a specific group of people, with either very high or very low stress levels, who participated or did not participate in the survey.

Thirdly, the SARS-CoV-2 pandemic may have acted as a confounding factor that influenced the levels and characteristics of MD in our population. As evidence, healthcare providers who cared for COVID-19 patients showed a higher MD score in our analysis. This result is consonant with a recent survey among PICU workers that demonstrated a positive correlation between MD and the COVID-19 pandemic [53]. Nonetheless, we believe that the pandemic, being a part of daily healthcare working life routine, must be taken into consideration, both to have an accurate estimation of the actual Italian MD level and to plan support activities.

Finally, the instrument’s scoring system led to non-normal data distribution, complicating statistical analysis. On the one hand, it ensures that items that scored 0, either by frequency or by intensity, do not affect the total MD score, but on the other hand, it creates a disparity in the chances of occurrence of each result and a peak of “0”. Furthermore, the MDS-R rating system asks interviewees to indicate the intensity of disturbance of each situation, even those they have never experienced. The latter requires respondents to think hypothetically, and this, although not affecting their final MD score, may cause them confusion or frustration (Appendix A). Our findings match other studies, which also recorded respondents’ difficulties on how to answer the intensity question in the event of zero frequency, but which still choose to ask those without experience to answer “level of disturbance” hypothetically, rather than not respond at all [19,50] Similarly, we chose the same approach, as this allows interviewees to reflect on, and respond to, all items. We can suggest that a different scoring system with a clear threshold between “no MD” and “MD” could have been more suited to both statistical and functional levels. However, as the scoring system is universally recognized for all MD measurement instruments, we did not make such a radical change to our scale, since this would make the comparison between Italian results and the rest of the scientific literature more difficult.

## 6. Conclusions

Moral Distress is a challenging problem for Italian ICUs in both adult and pediatric settings. The study expands the field of application of the Italian MDS-R by developing and validating its pediatric version. As a consequence of the target population on which we tested the new scale, it represents a valid instrument for measuring MD among all Italian healthcare workers who care for critically ill children. Our data demonstrated higher MD in clinicians who care for critically ill children in adult ICUs and a significant difference between northern, central, and southern Italy. These results deserve to be further investigated through new studies on a more extensive and longitudinal scale. Additional evidence-based and methodologically rigorous research would be advantageous to better explore the phenomenon, its short and long-term effects, and control strategies to improve healthcare professionals’ safety and well-being within the country.

## Figures and Tables

**Figure 1 ijerph-19-03880-f001:**
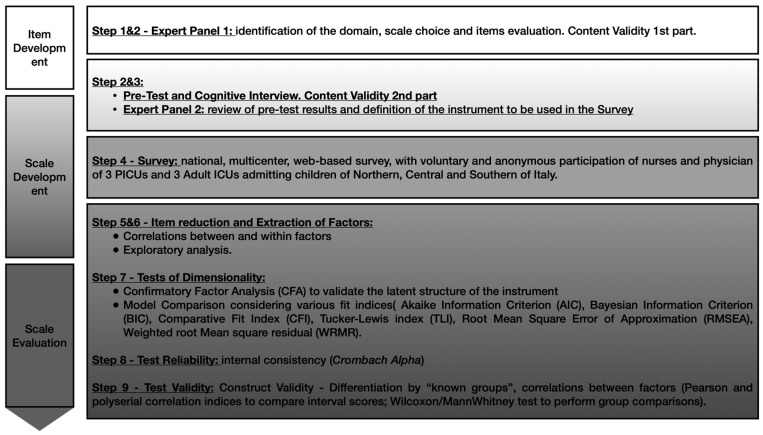
Study design. The study methodology follows the nine-step Boateng’s approach for developing and validating scales for health, social and behavioral research [18].

**Figure 2 ijerph-19-03880-f002:**
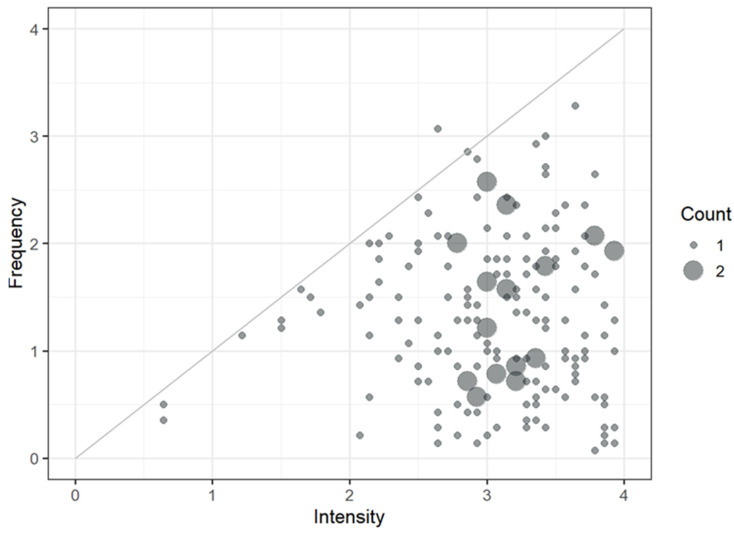
*Italian Pediatric MDS-R* results. Each subject’s overall intensity and frequency score were calculated, both as the average responses on the corresponding scale. The two resulting total scores were compared by placing them on a scatterplot where each dot represents a subject. When several subjects have the same value, the diameter of the corresponding dot increases in proportion to the number of subjects (Count). Almost all the points lay below the diagonal, mainly occupying the lower right quadrant highlighting that the intensities are generally high and the frequencies low.

**Figure 3 ijerph-19-03880-f003:**
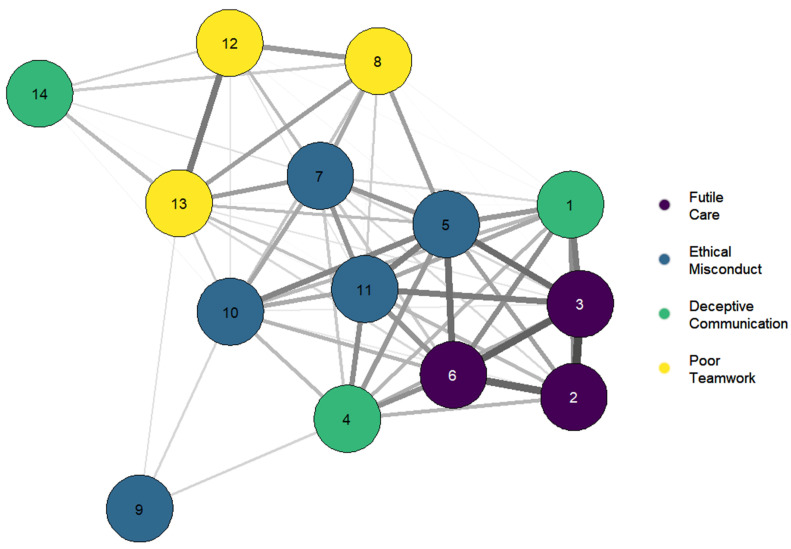
Network graph of the *Italian Pediatric MDS-R* factors. Each node represents an item whose color identifies the membership factor. The thickness of the connections is proportional to the size of the Pearson index (all correlations are positive). The bonds corresponding to correlations lower than 0.2 have been obscured to simplify the visualization. The most central nodes of the network correspond to items closely linked with others, while the peripheral nodes correspond to items with scarce and localized links.

**Figure 4 ijerph-19-03880-f004:**
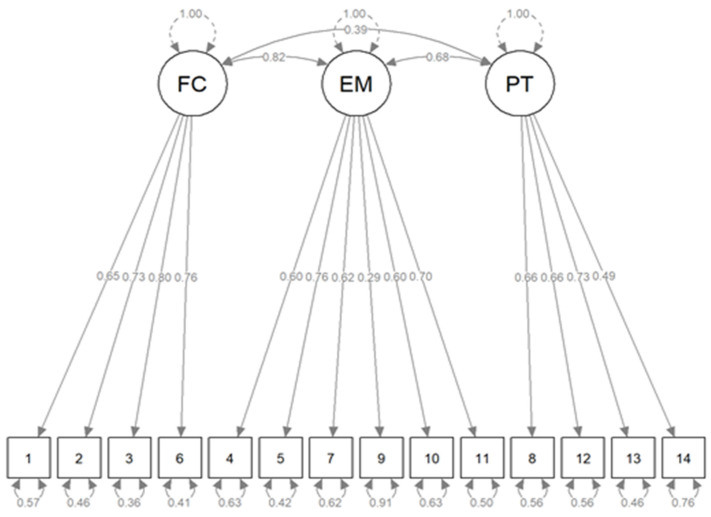
Path diagram of the Three-factor model (standardized solution). Identity numbers identify items. Factors: FC—Futile Care, EM—Ethical Misconduct, PT—Poor Teamwork.

**Figure 5 ijerph-19-03880-f005:**
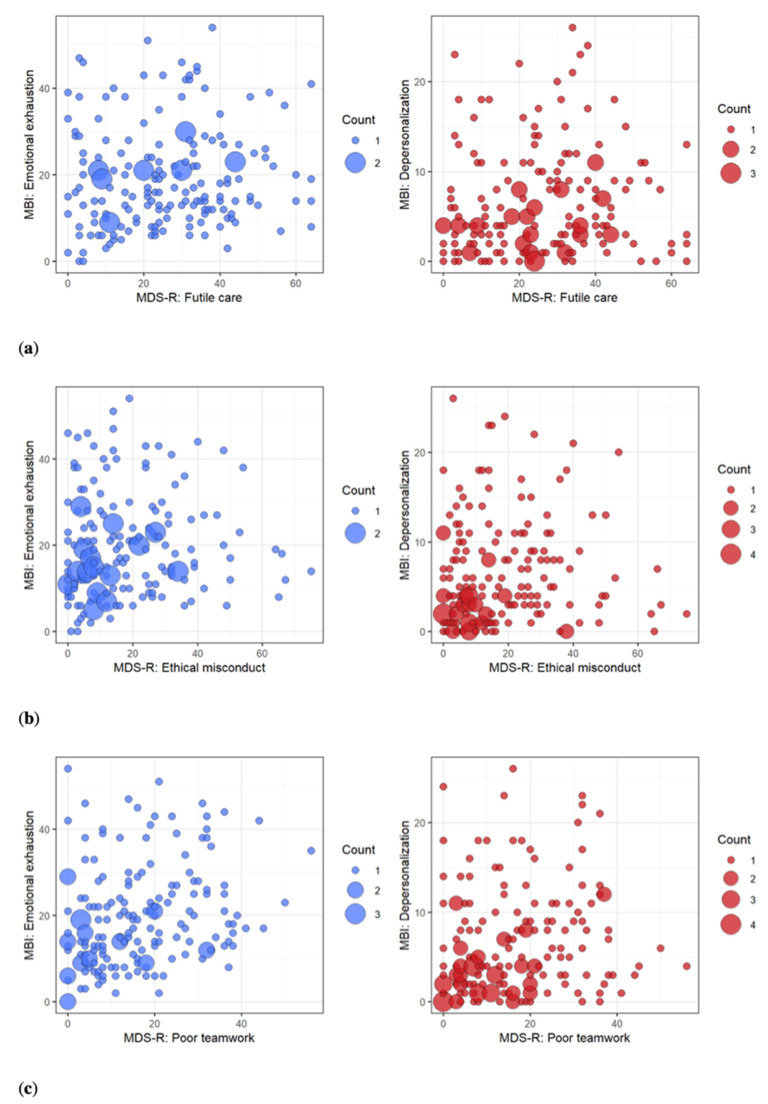
Visual inspection of the relationship between the factors of the *Italian Pediatric MDS-R* and the Maslach Burnout Inventory (MBI) sub-scales. The graph represents the relationships between the factors of the *Italian Pediatric MDS-R:* (**a**) Futile Care; (**b**) Ethical Misconduct; (**c**) Poor Teamwork and Emotional Exhaustion (EE) (**in blue**) and Depersonalization (DP) (**in red**) sub-scales of the MBI. Mainly referring to the relation with the MBI-DP scale, a fan-like structure can be observed with a correlation for low, but not for high, scores. With the limitations due to the small sample size and the resulting high variability, it seems possible to identify three groups: 1. low moral distress and no burnout; 2. high moral distress and low burnout; 3. high moral distress and high burnout.

**Table 1 ijerph-19-03880-t001:** Demographics. Demographics of participants in Pre-Test and Cognitive Interview (*n* = 14) and the web-based Survey (*n* = 182).

	Participants in the Cognitive Interview (14)	Participants in the Survey (182)
**Gender Identity, *n* (%)**		
Male	6 (42.9%)	57 (31.3%)
Female	8 (57.1%)	123 (67.6%)
**Age (years), median (min, max)**		
	41 (min 30, max 61)	40 (min 27; max 56)
**Role, *n* (%)**		
Physician	7 (50%)	66 (36.3%)
Nurse	7 (50%)	116 (63.7%)
**Unit, *n* (%)**		
PICU	8 (57.1%)	104 (57.1%)
Adult ICU	6 (42.9%)	78 (42.9%)
**Hospital Location within the Italian territory, *n* (%)**		
Northern	5 (35.7%)	97 (53.3%)
Central	4 (28.6%)	44 (24.2%)
Southern	5 (35.7%)	41 (22.5%)
**Years of experience with Critical Pediatric Patients, median (%)**		
3–11 months	1 (7.1%)	26 (14.3%)
1–4 years	2 (14.2%)	42 (23.1%)
5–9 years	3 (21.4%)	25 (13.7%)
10–19 years	4 (28.6%)	49 (26.9%)
20–29 years	3 (21.4%)	31 (17%)
≥30 years	1 (7.1%)	9 (4.9%)
***n*° of critically ill pediatric patients treated in the last year, *n* (%)**		
1–6	4 (28.57%)	58 (31.9%)
6–12	1 (7.14%)	17 (9.3%)
>12	9 (64.29%)	107 (58.8%)
**Having personally treated patients with SARS/CoV2 (COVID-19), *n* (%)**		
Yes	9 (64.3%)	146 (80.2%)
No	5 (35.7%)	36 (19.8%)

**Table 2 ijerph-19-03880-t002:** Descriptive statistics of the composite scores of the individual items of the *Italian Pediatric MDS-R*.

Item	Observed Range	Mean	Standard Deviation	Median	IQR
1. Witness healthcare providers giving “false hope” to parents.	0–16	5.44	4.49	4	6
2. Follow the family’s wishes to continue life support even though I believe that it is not in the best interest of the child.	0–16	6.88	5.01	6	9
3. Initiate extensive life-saving actions when I think that they only prolong death.	0–16	7.14	4.88	6	9
4. Follow the family’s request not to discuss death with a dying child who asks about dying.	0–16	2.23	3.58	0	4
5. Feel pressure from others to order what I consider to be unnecessary tests and treatments.	0–16	5.37	4.55	4	6
6. Continue to participate in care for a hopelessly ill child who is being sustained on a ventilator when no one will make a decision to withdraw support.	0–16	6.12	5.21	6	8
7. Avoid taking action when I learn that a physician or nurse colleague has made a medical error and does not report it.	0–16	2.66	3.38	2	4
8. Work with a physician or a nurse who, in my opinion, is providing incompetent care.	0–16	4.35	3.91	4	6
9. Increase the dose of sedatives/opiates for an unconscious child that I believe could hasten the child’s death.	0–16	1.73	2.74	0	3
10. Take no action about an observed ethical issue because the	0–16	2.36	3.71	0	4
involved staff members or someone in a position of authority requested that I do nothing.					
11. Follow the family’s wishes for the child’s care when I do not agree with them but do so because of fears of a lawsuit.	0–16	3.57	4.6	2	6
12. Watch patient care suffer because of a lack of provider continuity.	0–16	4	4.52	3	8
13. Witness diminished patient care quality due to poor team communication.	0–16	5.55	4.55	4	5
14. Ignore situations in which parents have not been given adequate information to ensure informed consent.	0–16	2.36	3.28	0	4

**Table 3 ijerph-19-03880-t003:** Descriptive statistics for relevant socio-demographic groups.

Group	Size	Mean	Standard Deviation	Median	IQR
Females	119	65.39	36.21	64	50.5
Males	57	47.65	32.84	45	41
Nurses	113	64.29	35.94	58	52
Physicians	64	51.77	34.87	45	52.25

## Data Availability

Data supporting reported results can be requested to the corresponding author, chiaragrasso@protonmail.com. There are not publicly archived datasets analyzed or generated during the study.

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
