# Peer review of "Moral Distress in Healthcare Providers Who Take Care of Critical Pediatric Patients throughout Italy—Cultural Adaptation and Validation of the Italian Pediatric Instrument"

_ijerph, 2022, doi:10.3390/ijerph19073880_

Round 1

Reviewer 1 Report

Dear authors,

This is a very interesting manuscript, I congratulate you for your hard work. As suggestions:

  • the abbreviation MDS-R appear in the Abstract; please explain it here also - line 37 and 46
  • In the introduction chapter I would like to read more about the adverse effects of MD on medical staff 
  • which was the way you chose the 7 nurses and the 7 physiscians for the Cognitive interviewing?
  • the Statistical analysis chapter and the Results chapter are too technical and hard to follow
  • Tabel 1, Demographics, is overloaded with data and difficult to track; maybe it should be structured differently
  • The Discussions chapter should include more comparisons with data from the literature, perhaps with similar scales used in other countries.

Author Response

Dear reviewer, 

We are glad you found our paper interesting. Here are the revisions/answers to your suggestions:

  • the abbreviation MDS-R appear in the Abstract; please explain it here also - line 37 and 46 DONE
  • In the introduction chapter I would like to read more about the adverse effects of MD on medical staff We reviewed and broadened the introduction. We added a further description of the MD adverse effects. Line 100-110.
  • which was the way you chose the 7 nurses and the 7 physiscians for the Cognitive interviewing? We better explained the sampling criteria we used in chapter 2.3 line 807-811
  • the Statistical analysis chapter and the Results chapter are too technical and hard to follow. We understand it may appear too technical. However, we think this is a strength and a value of our research. We strongly believe it is mandatory and essential to first validate an instrument with in-depth analysis to obtain a reliable and valid measurement instrument. We think it is possible to draw further and more clinical research on the same field only with these basics. 
  • Tabel 1, Demographics, is overloaded with data and difficult to track; maybe it should be structured differently We simplified the table.
  • The Discussions chapter should include more comparisons with data from the literature, perhaps with similar scales used in other countries. We modify the discussion by adding further comparison. However unfortunately there are not so many validation studies that performed an in-depth analysis. On the one hand, this represents the value of our paper; on the other hand, it makes the comparison more difficult. 

Reviewer 2 Report

I am grateful to the editor for providing me with this opportunity to be among the first readers of this draft entitled “Moral Distress in the Healthcare Providers who take care of Critical Pediatric Patients throughout Italy. MoDiPerSaPerCI. Cultural adaptation and validation of the Italian Pediatric instrument”

I have read the efforts of the authors and found that the authors have initiated a useful academic debate. The paper is well written, I did not find any critical issue in publishing this work.

There are few suggestions that I would mention as under

Abstract: Please add motivation statement in the abstract, instead of objective statement. Please ensure that the abstract has the following elements: 1-2 sentences on the context and the need for the study; 1-2 sentences on the methodology; the majority of the abstract on the actual results of the study; 1-2 sentences on key conclusions and recommendations

  • Better frame the knowledge gap that you address, and make a clear statement of         
  • Clean up the argumentation line, and provide theoretical justification of the choices for your study.
  • Elaborate better the last section of your study - limitations and future research.
  • Please add more literature in the introduction section.
  • Please clearly explain your contribution of the study
  • Better to cite the latest and most relevant studies from a similar topic.
  • Update the paper with recent references
  • Practical implications: Please add a section in the discussion section to discuss the implications of the findings for policymakers and other relevant stakeholders.
  • Please separate limitation section from discussion and conclusion.
  • Please focus on key findings and summaries key recommendations to stakeholders and for future research.
  • The conclusion is stated in a general form, for a text to be published in an esteemed journal (like this), it is necessary to conclude your research in more convincing way
  • Please remove grammatical error

Other concerns

Introduction:

I suggest finding a knowledge gap in a scientific paper that is no more than five years old published in an indexed scientific journal, where a relevant author invites to advance the knowledge gap. Such a knowledge gap should be included in the form of a sentence in quotation marks, verbatim, in the introduction section, and will be the rudder that will guide your research. Please explain well what the message of your paper is.

Conclusion:

I suggest that in the conclusion section, you explain how the results of your research fill the knowledge gap you found

Author Response

Dear reviewer,

We are glad You found our paper valuable and useful. 

Please find here our revisions/answers to your suggestions:

Abstract: Please add motivation statement in the abstract, instead of objective statement. Please ensure that the abstract has the following elements: 1-2 sentences on the context and the need for the study; 1-2 sentences on the methodology; the majority of the abstract on the actual results of the study; 1-2 sentences on key conclusions and recommendations DONE. We modified the abstract. We followed the journal indications of the note for the authors.

  • Better frame the knowledge gap that you address, and make a clear statement of   DONE. Please see lines 33-36 and 122-131.      
  • Clean up the argumentation line, and provide theoretical justification of the choices for your study. DONE. Please see lines 33-36 and 122-131.
  • Elaborate better the last section of your study - limitations and future research. DONE. We rewrote and modified the discussion and limitations section.
  • Please add more literature in the introduction section. DONE.
  • Please clearly explain your contribution of the study. DONE.
  • Better to cite the latest and most relevant studies from a similar topic. DONE.
  • Update the paper with recent references. DONE.
  • Practical implications: Please add a section in the discussion section to discuss the implications of the findings for policymakers and other relevant stakeholders. DONE. We rewrote and modified the discussion and limitations section.
  • Please separate limitation section from discussion and conclusion. DONE.
  • Please focus on key findings and summaries key recommendations to stakeholders and for future research. DONE. We rewrote and modified the discussion and limitations section.
  • The conclusion is stated in a general form, for a text to be published in an esteemed journal (like this), it is necessary to conclude your research in more convincing way. DONE. Please check line 1976 - 1987
  • Please remove grammatical error. DONE. We provided an in-depth English review.

Other concerns

Introduction:

I suggest finding a knowledge gap in a scientific paper that is no more than five years old published in an indexed scientific journal, where a relevant author invites to advance the knowledge gap. Such a knowledge gap should be included in the form of a sentence in quotation marks, verbatim, in the introduction section, and will be the rudder that will guide your research. Please explain well what the message of your paper is. DONE. Please see lines 33-36 and 122-131.

Conclusion:

I suggest that in the conclusion section, you explain how the results of your research fill the knowledge gap you found DONE. Please check line 1976 - 1987

Thank You for Your time and consideration.

Best Regards.

Reviewer 3 Report

Important, well designed, conducted, and presented study

Minor specific comments

Title - Delete acronym "MoDiPerSaPerCI."

Table 1 - for better readability it should be distended to the whole page width

Line 493 - Data is plural, thus the sentence should read "... data show...", doublecheck for whole manuscript

Thank you

Author Response

Dear reviewer,

We are glad you appreciated our research.

Please find here the revisions/answers to Your suggestions:

Minor specific comments

Title - Delete acronym "MoDiPerSaPerCI." DONE

Table 1 - for better readability it should be distended to the whole page width. We shortened and simplified table 1. We cannot modify the layout as the editor defines it.

Line 493 - Data is plural, thus the sentence should read "... data show...", doublecheck for whole manuscript. DONE thank you.

Thank you for Your time and consideration.

Best Regards

Reviewer 4 Report

The reviewed paper introduces a pediatric version of the Italian Moral Distress Scale-Revised. Hence, it is a typical psychometric paper providing an adaptation of an existing measure. The paper itself seems valuable, particularly due to the utility of the scale for clinical practice and research. Nevertheless, some issues should be definitely addressed before the paper is suitable for publication. I list them below.

  1. Although I am not a native English speaker, I had some serious doubts regarding language correctness in some parts of the manuscript. I will leave it to the assessment of the paper’s editor, however I believe that the paper could benefit from a thorough proof-reading by a native English speaker (e.g., “small number of sample”).
  2. Despite my suggestion raised in p. 1, I believe that the paper is concisely written and communicative. Its novelty is limited due to the fact that the manuscript actually introduced an adaptation of an already existing scale. Nevertheless, I believe that such publications are necessary to provide an evidence for a validity of psychometric measurement, hence I assess the utility of the Italian language version of the paper as high.
  3. I would strongly recommend to provide a reader with a clear definition of moral distress, as measured by the IP MDS-R, accompanied with established external validity indicators and a nomological network (this would probably justify the selection of the selected validity indicators).
  4. I am not sure what exactly did the authors want to say by stating “The meeting focused on confirming the instrument domain together with the lack of existing instrument to measure the MD in the Italian Pediatric Critical Care setting.” (lines 121-122). The entire paragraph 2.2 should be carefully re-read and revised, in order to provide a clear description of this stage of work on the adaptation.
  5. If the score for each item is computed via multiplying frequency X intensity, then why are the respondents asked “to rate the intensity of the disturbance that would cause that specific situation in case it occurs”? Yet in such a situation the ultimate score for the item is zero, regardless of the answer. This should be clarified.
  6. I have doubts regarding assigning each of the items initially comprising the Deceptive Communication factor. I see that this relocation is at least partly justified by the factor structure reflected in the 3F model, however I am not convinced whether the content of each of the items indeed fits the definition of the factor to which the Authors assigned it. Also the discussion of this issue provided in lines 459-476 does not sound too convincing. Finally, only the relocation of item 14 is discussed – what about the justification behind two analogical decisions for the two remaining DC items? I would expect a critical reflection and discussion of the actual conceptual match between the three factors and the items added to each of them. Also, the Authors could consider limiting the questionnaire to the items originally comprising the three factors, thus neglecting the poorly fitted fourth factor instead of merging its items with the three remaining scales.
  7. Minor issue: In figure 3, please provide the full names of the scales/dimensions. Each figure should be readable without the need to refer to the text.
  8. The lack of significant associations with validity criteria (SWLS, burnout, etc.) seems a bit confusing to me. In the light of these (negative) results, the claim that IP MDS-R is a valid and reliable measure does not seem to have a strong support in the data presented in the paper. These claims should be smoothed. 
  9. The Authors should specify which estimation method did they use. Most commonly used methods include weighted least squares (WLS), weighted least squares mean-andvariance-adjusted (WLSMV), unweighted least squares mean-and-variance adjusted (ULSMV), maximum likelihood (ML), robust maximum likelihood (MLR) and Bayesian estimation methods. Please provide the information accompanied with a justification of the choice.

Author Response

Dear reviewer,

We are glad You find our research valuable. 

Please find here our revisions/answers to Your suggestions:

  1. Although I am not a native English speaker, I had some serious doubts regarding language correctness in some parts of the manuscript. I will leave it to the assessment of the paper’s editor, however I believe that the paper could benefit from a thorough proof-reading by a native English speaker (e.g., “small number of sample”). DONE. We provided an in-depth language revision.
  2. Despite my suggestion raised in p. 1, I believe that the paper is concisely written and communicative. Its novelty is limited due to the fact that the manuscript actually introduced an adaptation of an already existing scale. Nevertheless, I believe that such publications are necessary to provide an evidence for a validity of psychometric measurement, hence I assess the utility of the Italian language version of the paper as high. Thanks.
  3. I would strongly recommend to provide a reader with a clear definition of moral distress, as measured by the IP MDS-R, accompanied with established external validity indicators and a nomological network (this would probably justify the selection of the selected validity indicators). DONE. We broadened the introduction, and we added a chapter where we clearly stated the indicators of validity we used. Please see chapter 2.6, lines 914-923.
  4. I am not sure what exactly did the authors want to say by stating “The meeting focused on confirming the instrument domain together with the lack of existing instrument to measure the MD in the Italian Pediatric Critical Care setting.” (lines 121-122). The entire paragraph 2.2 should be carefully re-read and revised, in order to provide a clear description of this stage of work on the adaptation. We changed and rewrote all the chapters.
  5. If the score for each item is computed via multiplying frequency X intensity, then why are the respondents asked “to rate the intensity of the disturbance that would cause that specific situation in case it occurs”? Yet in such a situation the ultimate score for the item is zero, regardless of the answer. This should be clarified. Unfortunately, the original MDS scoring system contemplates this characteristic that is universally recognized. We just followed the general rule. Nonetheless, we added an argumentation of this scoring system's trait in the discussion Line 1953-1966. But we do not intend to radically change our scale since this would make the comparison between Italian results and the rest of the scientific literature more difficult.  
  6. I have doubts regarding assigning each of the items initially comprising the Deceptive Communication factor. I see that this relocation is at least partly justified by the factor structure reflected in the 3F model, however I am not convinced whether the content of each of the items indeed fits the definition of the factor to which the Authors assigned it. Also the discussion of this issue provided in lines 459-476 does not sound too convincing. Finally, only the relocation of item 14 is discussed – what about the justification behind two analogical decisions for the two remaining DC items? I would expect a critical reflection and discussion of the actual conceptual match between the three factors and the items added to each of them. Also, the Authors could consider limiting the questionnaire to the items originally comprising the three factors, thus neglecting the poorly fitted fourth factor instead of merging its items with the three remaining scales.

    The "destiny" of the DC scale is undoubtedly a point requiring further in-depth analyses. The present paper will direct future investigations on a comprehensive and more representative sample. For this reason, we preferred not to exclude the items of the DC scale, waiting for a second stage of the study. For what concerns items 1 and 4, we expanded the discussion.

  7. Minor issue: In figure 3, please provide the full names of the scales/dimensions. Each figure should be readable without the need to refer to the text. DONE.
  8. The lack of significant associations with validity criteria (SWLS, burnout, etc.) seems a bit confusing to me. In the light of these (negative) results, the claim that IP MDS-R is a valid and reliable measure does not seem to have a strong support in the data presented in the paper. These claims should be smoothed. We tried to clarify this point in the discussion. Also, we added a specific chapter (2.6) with the hypotheses we chose as proof of the validity of the new scale.  We consider MD a psychological construct subject to high variability; we believe that MD score-burnout relationship in our target population deems to be deepened in further research with a larger sample size before drawing some conclusions.
  9. The Authors should specify which estimation method did they use. Most commonly used methods include weighted least squares (WLS), weighted least squares mean-andvariance-adjusted (WLSMV), unweighted least squares mean-and-variance adjusted (ULSMV), maximum likelihood (ML), robust maximum likelihood (MLR) and Bayesian estimation methods. Please provide the information accompanied with a justification of the choice. 

    Please note that we had briefly discussed this issue at the end of section 2.7.

Thank You for Your time and consideration.

Best Regards

Reviewer 5 Report

In this manuscript, Grasso et al develop and validate the pediatric version of the Italian Moral Distress Scale-Revised.

The main finding of the study is The Italian Pediatric MDS-R is a valid and reliable instrument for measuring MD among the Italian health-workers who care for critically ill children. The manuscript is well studied and a few minor corrections below will help to improve the quality of the manuscript.

Minor comments:

  1. Methods, provide in detail (not in present form) a flow chart that summarizes the clinical design and data analysis
  2. Table 1, providing a simple representative is more easy and effective for the readers.
  3. Discussion is should be elaborate and discuss about present scenario.

Author Response

Dear reviewer,

We are glad You found our paper valuable. Please see here our revisions/answers to Your suggestions:

Minor comments:

  1. Methods, provide in detail (not in present form) a flow chart that summarizes the clinical design and data analysis DONE. We modified figure 1—study design.
  2. Table 1, providing a simple representative is more accessible and effective for the readers. DONE. We simplified table 1.
  3. Discussion is should be elaborate and discuss the present scenario. DONE. We revised and rewrote the debate. We added recent papers on the same topic and the current scenario.

Thank You for Your time and consideration.

Best Regards.

Round 2

Reviewer 1 Report

Dear Authors,

I believe that the revised version of the manuscript meets all the conditions for publication. Congratulations for your work.